# Clinical Results of the Use of Low-Cost TKA Prosthesis in Low Budget Countries—A Narrative Review

Edoardo Bori [ID], Clara Deslypere, Laura Estaire Muñoz and Bernardo Innocenti *[ID]

BEAMS Department, Université Libre de Bruxelles, 1050 Bruxelles, Belgium; edoardo.bori@gmail.com (E.B.)
* Correspondence: bernardo.innocenti@ulb.be

**Abstract:** Despite the orthopedics markets in the US and the EU reaching a plateau, the market size in countries such as Brazil, Russia, India, and China is steadily growing. As a result, major orthopedic companies are shifting their focus towards these markets and developing products tailored to their needs. However, a significant challenge associated with this new opportunity is the requirement for the development of more affordable prostheses compared to those sold in the US and Europe. With the introduction of these lower-cost models into the market, this article aims to assess their performance in comparison to traditional models. A literature review was conducted, analyzing four parameters—the Hospital for Special Surgery Score, Knee Society Score, Range of Motion, and Western Ontario and McMaster Universities Arthritis Index—to evaluate different models. The findings indicated that low-cost models perform either equally well or, in some cases, slightly worse than traditional ones. It is worth to mention that the existing literature on this topic is limited, resulting in a relatively small number of models and studies included in this specific study. Nevertheless, this latter serves as a valuable foundation for future in-depth analyses and investigations.

**Keywords:** TKA; BRICS countries; low-budget; scores





## 1. Introduction

In recent years, with the saturation of the orthopedic device market in the United States and Europe, major companies in the field have shifted their focus to explore new potential markets. One such promising market is the low- and middle-income countries, which have a combined population of over 6.1 billion [1]. Within these countries, the orthopedic market size is estimated to be over 30.5 million people, considering that approximately 0.5% of the population in developing countries requires orthopedic devices [2].

Among the range of low- and middle-income countries, the BRICS countries (Brazil, Russia, India, China and South Africa) present significant opportunities. These four countries alone account for 37% of the world's population [1] and have economies growing at a faster pace than that of the United States, making them the largest medical market outside of the US, Europe, and Japan. Additionally, there is a growing middle class in these countries that not only demands higher-quality services but also specifically seeks better healthcare. For instance, in India, the middle class is projected to grow from 80 million people today to 580 million by 2025, representing 41% of the population [3]. Therefore, these countries offer a substantial and foreseeable market for major orthopedic companies like Stryker (Kalamazoo, MI, USA), DePuy (Warsaw, IN, USA), Zimmer Biomet (Warsaw, IN, USA), Smith and Nephew (London, UK), and Medtronics (Dublin, Ireland), especially as the EU and US markets become more challenging.

However, entering this new market also presents certain challenges. The average GDP per capita in low- and middle-income countries is approximately $4417.8 [1]. In terms of healthcare services, nearly 50% of the financing comes from out-of-pocket payments, while only 38% comes from combined funding pools in these countries. In comparison, high-income countries rely on combined funding pools for 80% of healthcare financing [4].

Consequently, the purchasing power for medical devices in these countries is significantly lower than in traditional orthopedic markets such as the US and the EU.

Furthermore, some of these countries impose restrictions on the pricing of orthopedic devices. For example, in 2019, the Indian government reduced the allowed cost for total knee arthroplasty (TKA) surgery by 59% to 69%, lowering the price from the previous amount of Rs 158,324 (approximately $2200) to Rs 54,720 (around $750) [5]. These price limitations further impact the profitability and market dynamics for orthopedic companies operating in these regions. On the other hand, the prices adopted by the big Orthos (as of 2011) stood between $7775 and $12,495 for Smith and Nephew, between $7470 and $16,510 for Zimmer Biomet, between $8514 and $16,743 for DePuy and between $5704 and $15,662 for Stryker [6]. Needless to say, these budgets are not sustainable in developing countries; the big companies were therefore presented with the challenge of providing dedicated models to be able to cover also this section of the market.

To tackle this challenge, large and medium-sized companies adopted two different strategies. Some companies developed an in-house dedicated low-cost model, while others opted to acquire companies that produce low-cost devices to gain access to their products and customer base. Smith and Nephew pursued the first approach, resulting in the development of the Anthem TKA. On the other hand, Medtronic followed the second approach by acquiring Responsive Orthopaedics (Minneapolis, MN, USA) in 2016 [7], and Meril Life (Vapi, Daman and Diu, India) purchased Maxx Ortho (Norristown, PA, USA) in 2009 [8]. In addition to these major players, there are also smaller orthopedic companies operating in the low-cost market, such as Baumer (São Paulo, Brazil).

The commercialization of these low-cost prostheses has sparked investigations into comparing their performance with traditional ones, raising questions about the price-quality gap. In order to address these questions, this article examines the performance of selected low-cost and traditional prostheses by analyzing measured clinical outcomes from various research studies.

## 2. Materials and Methods

A literature review was conducted to examine the existing prostheses available in low budget countries. The initial hypothesis was based on the observation that low-cost models were widely sold in these countries, particularly in India, due to price constraints. The study focused solely on primary knee prostheses to narrow down the scope, and several models meeting these criteria were identified. Among them, the Baumer AKS model and the Maxx Ortho Freedom model were selected, based on the availability of clinical study results for comparison. For these selected models, studies referencing the Range of Motion (ROM), the Hospital for Special Surgery (HSS) score, and the Western Ontario and McMaster Universities Arthritis Index (WOMAC) were found, and these three scores were chosen as means of comparison between the different models.

The Range of Motion (ROM) is calculated by summing the maximum flexion angle and the maximum extension angle of the knee. The HSS score is a total score out of 100, divided into seven categories, including pain, function, range of motion, muscle strength, flexion deformity, instability, and subtractions (where a higher score indicates better outcomes). The score is based on a combination of patient interviews (50%) and physical examinations (50%) [9].

The Knee Society Score (KSS) complements the HSS score by incorporating an evaluation of instability in the anteroposterior plane and a classification system for patients with associated medical conditions. It consists of the Knee Score (100 points), the Knee Function Score (100 points), and a patient classification system, assessing pain relief, range of motion, stability, ability to walk, and ability to go up and down stairs [9].

The WOMAC index is a widely used evaluation tool for Hip and Knee Osteoarthritis. It comprises 24 self-administered questionnaire items divided into three subscales: pain, stiffness, and physical function. To facilitate comprehension and analysis of the results, the studies inverted the Likert scale used in the original questionnaire. In this inverted

scale, "1" represents the worst result, and "5" represents the best result (in the original questionnaire, the best result was 0, and the worst result was 4). This modification was made to align with the Likert scale commonly used in questionnaires and opinion polls, as explained by Likert himself in a published report [10].

To compare the results of these low-cost models with traditional prostheses, the best-selling TKA models from Smith and Nephew, Zimmer Biomet, Stryker, and DePuy were chosen: the Legion, Persona, Triathlon, and Attune models, respectively. These selections were based on their popularity in Australia, where the Legion ranked as the 8th most used cemented TKA, the Persona ranked as the 4th most used cementless TKA and 2nd cemented TKA, the Triathlon was the most used cementless and cemented TKA, and the Attune ranked as the 3rd most used cemented TKA [11].

A literature review was conducted to gather information on these selected traditional prostheses. Only studies that provided data on ROM, HSS, WOMAC, or KSS were considered. As not all studies reported data for all models, the comparison pool was expanded to include the KSS. With these four parameters, a performance comparison of the chosen models was conducted.

## 3. Results

### 3.1. Study Characteristics

The length of the studies ranges from 0.5 to 6.5 years with a mean of 2.3 years (standard deviation 1.58). The sample size of each study varies between 21 et 2656 replaced knees with a mean of 922 knees (Table 1).

**Table 1.** Study characteristics of the studies included.

| Model | Study | Sample Size (Knees) | Mean Age or Range | % Female | Length of Follow-Up (Years) |
|---|---|---|---|---|---|
| **Freedom** | **Weighted Mean (SD)** | **183.8 (7.50)** | **69.7 (0.02)** | **65.6% (9.00%)** | **5.4 (1.10)** |
| | Durbhakula et al., 2019 [12] | 176 | 69.7 | 75% | 6.5 |
| | Singh et al., 2017 [13] | 191 | 67.67 | 57% | 4.3 |
| **Persona** | **Weighted Mean (SD)** | **129.2 (33.15)** | **68.0 (1.31)** | **65.1% (33.82%)** | **2.5 (0.54)** |
| | Ryu et al., 2020 [14] | 143 | 69.5 | 85% | 3.1 |
| | Kim et al., 2019 [15] | 143 | 66.7 | NR | 2 |
| | Indelli et al., 2018 [16] | 50 | 67.6 | 8% | 2 |
| **Triathlon** | **Weighted Mean (SD)** | **1793.3 (593.22)** | **65.6 (1.53)** | **61.4% (0.91%)** | **1.8 (0.09)** |
| | Palmer et al., 2014 [17] | 338 | 69.4 | 63.6% | 2 |
| | Harwin et al., 2008 [18] | 2035 | 65 | 61% | 1.75 |
| **AKS** | **Weighted Mean (SD)** | **90.1 (22.62)** | **68 (NE)** | **73% (NE)** | **0.9 (0.48)** |
| | Souza et al., 2014 [19] | 107 | 68 | 73% | 0.5 |
| | Schwartsmann et al., 2017 [20] | 60 | (60–80) | NR | 1.5 |
| **Legion** | **Weighted Mean (SD)** | **2563.3 (478.05)** | **68.4 (NE)** | **63.9% (NE)** | **3.4 (0.56)** |
| | Saccone et al., 2011 [21] | 2656 | 68.4 | 63.9% | 3.5 |
| | Chow et al., 2016 [22] | 100 | (18–99) | NR | 0.5 |
| **Attune** | **Weighted Mean (SD)** | **59.6 (24.42)** | **NE (NE)** | **73% (NE)** | **1.80 (0.54)** |
| | Kaptein et al., 2020 [23] | 38 | (21–90) | 73% | 2 |
| | Hauer et al., 2020 [24] | 80 | >60 | NR | 2 |
| | Carey et al., 2018 [25] | 21 | NR | NR | 0.5 |

NR = Not Reported. NE = Not Evaluated.

The number of implants for each model is the following: 367 Freedom, 336 Persona, 2373 Triathlon, 167 AKS, 2756 Legion and 139 Attune implants.

### 3.2. Revision Rates

In the analyzed studies, the Freedom model has revision rates ranging from 0 to 1.7%. The Persona has revision rates ranging from 1.9 to 3.99%. The Triathlon's revision rates range from 2.1% (for the CR model) to 6.1% (for the PS model). The Legion's revision rates range from 3.6% (for the CR model) to 3.9% (for the PS model). The revision rates of the Attune range from 1.5% to 2.5%. No information reported about the AKS model revision rates in the studies reviewed (Table 2).

**Table 2.** Estimated Revision Rate.

| Model | Study | Revision Rate | | | | | | | | | | |
|---|---|---|---|---|---|---|---|---|---|---|---|---|
| | | After Surgery | 1 Year | 2 Years | 3 Years | 4 Years | 5 Years | 6 Years | 7 Years | 8 Years | 9 Years | 10 Years |
| **Freedom** | Durbhakula et al., 2019 [12] | 1.70 | | | | | | | | | | |
| | Singh et al., 2017 [13] | | | | 0 | | | | | | | |
| **Persona** | Ryu et al., 2020 [14] | 1.80 | | 3.50 | | | | | | | | |
| | Kim et al., 2019 [15] | | | | | | NR | | | | | |
| | Indelli et al., 2018 [16] | 2.00 | | | | | | | | | | |
| | Australian Registry, 2020 [11] | | | | | | 1.90 | | | | | |
| | NJR, 2019 [26] | | | | | | 3.99 | | | | | |
| **Triathlon** | Australian Registry, 2020 [11] | | | | | | | | | | | 2.10 (CR) 6.10 (PS) |
| | NJR, 2019 [26] | | | | | | | | | | | 3.89 |
| **Legion** | Australian Registry, 2020 [11] | | | | | | 3.60 (CR) 3.90 (PS) | | | | | |
| **Attune** | Michigan Registry, [27] | | 0.74 | 1.61 | 2.50 | | | | | | | |
| | UK Registry, 2019 [28] | | 0.40 | 1.00 | 1.50 | 1.80 | 2.20 | 2.30 | 2.30 | | | |
| | Australian Registry, 2020 [29] | | 0.90 | 1.50 | | | | | | | | |

NR = not reported, CR = Cruciate Retaining, PS = Posterior Stabilized.

### 3.3. Explanation of Comparison Parameters

The clinical parameters taken into account were the ROM, the HSS score, the KSS and the WOMAC.

#### 3.3.1. Comparison of Range of Motion

The pre-operative ROM ranges from 93.4° to 114° and the post-op ROM ranges from 107° to 130.3° (Table 3).

The mean improvement of the ROM is 15.37° for the Freedom (13.5% of improvement), 16.9° for the Persona (15.6% of improvement), 19° for the Triathlon (17% of improvement), 10° for the Legion (8.7% of improvement) and 16.3° for the Attune (21% of improvement) (Figure 1).

**Table 3.** Range of Motion Pre- and Post-operative.

| Model | Study | Follow-Up Time | Average Pre-Op ROM | ROM after Follow-Up Time | *p*-Value |
|---|---|---|---|---|---|
| **Freedom** | Durbhakula et al., 2019 [12] | 5 years | 113.8 ± 6.1 | 128.7 ± 4.1 | <0.001 |
| | Singh et al., 2017 [13] | 3 years | 104 ± 5.67° (range, 85°–119°) | 119.8 ± 11.05° (98°–123°) | <0.05 |
| **Persona** | Ryu et al., 2020 [14] | 3.1 years | 108.4° (70 to 145°) | 130.3° (105 to 150°) | <0.0001 |
| | Kim et al., 2019 [15] | 2 years | 105.3 ± 8.7 | 126.1° ± 10.8° (range 95°–140°) | <0.05 |
| | Indelli et al., 2018 [16] | 2 years | 112° | 120° (CI 3.8) | NR |
| **Triathlon** | Harwin et al., 2008 [18] | Post-operatively | 109 | 128 | NR |
| **Legion** | Saccone et al., 2011 [21] | 2 years | 114 | 124 | 0.02 |
| **Attune** | Hauer et al., 2020 [24] | 2 years | 93.4 ± 21.8 | 113.0 ± 10.6 | NR |
| | Carey, et al., 2018 [25] | 0.5 years | 94 | 107 | |

NR = not reported.

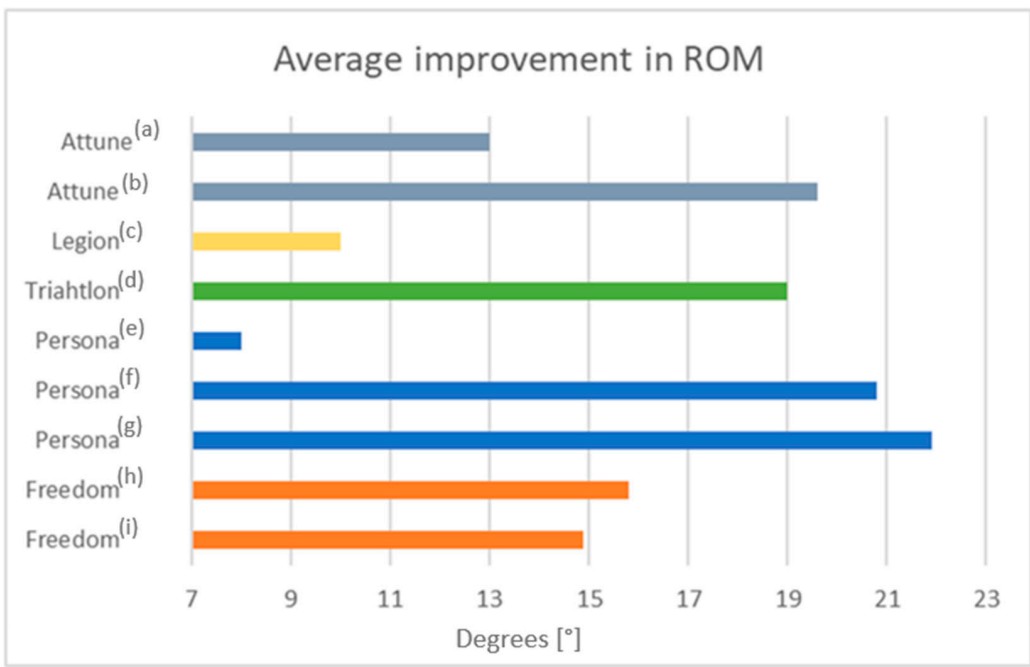

**Figure 1.** Average improvement in ROM. Each color represents a specific TKA model, i.e., Attune (gray, a [5], b [14]), Legion (yellow, c [27]), Triathlon (green, d [13]), Persona (blue, e [17], f [19], g [26]), and Freedom (orange, h [29], i [11]).

No information reported about the AKS model range of motion in the studies reviewed (Table 3).

3.3.2. Comparison of KSS

The pre-operative KSKS ranges from 40.9 to 74 and the post-op KSKS ranges from 88.36 to 97.5. The mean improvement of the KSKS value is 20 for the Persona, 47.85 for the Triathlon, 51.75 for the Legion and 40.5 for the Attune. And the post-op KSKS value for the ASK model is 88.36 ± 9.64 (patellar resurfaced) and 84.26 ± 9.44 (non-patellar resurfaced). The pre-operative KSFS ranges from 38 to 74 and the post-op KSFS ranges from 75 to 93.5. The mean improvement of the KSFS value is 17 for the Persona, 21.7 for the Triathlon, 38.5 for the Legion and 41 for the Attune. No information reported about the Freedom model KSS values in the studies reviewed (Table 4).

**Table 4.** Knee society score (KSS) Pre- and Post-operative.

| Model | Study | Average Follow-Up Time | Average Pre-Op KSKS Score | KSKS Score after Follow-Up Time | Mean Change in KSK | Mean Percentual of Improval in KSKS (%) | Average Pre-Op KSFS Score | KSFS Score after Follow-Up Time | Mean Change in KSFS | Mean Percentual of Improval in KSFS (%) |
|---|---|---|---|---|---|---|---|---|---|---|
| **AKS** | Souza et al., 2014 [19] | 6 months | NR | PR: 88.36 ± 9.64. NPR: 84.26 ± 9.44 | NR | NR | NR | NR | NR | NR |
| **Persona** | Kim et al., 2019 [15] | 2 years | 74.0 | 94.00 | 20.0 | 27.00 | 74.0 | 91.0 | 17.0 | 22.97 |
| **Triathlon** | Palmer et al., 2014 [17] | 2 years | 41.8 | 89.50 | 47.7 | 114.10 | 49.7 | 71.7 | 21.4 | 44.27 |
| | Harwin et al., 2008 [18] | Post-operatively | 48.0 | 96.00 | 48.0 | 100.00 | 63.0 | 85.0 | 22.0 | 34.92 |
| **Legion** | Saccone et al., 2011 [21] | Post-operatively | 43.1 | 90.60 | 46.9 | 110.00 | NR | NR | NR | NR |
| | Chow J. et al., 2016 [22] | 6 months | NR | NR | NR | NR | 38.0 | 75.0 | 37.0 | 97.37 |
| | Brown RB. et al., 2019 [30] | 5 years follow up | 40.9 | 97.50 | 56.6 | 138.39 | 53.9 | 93.5 | 40.0 | 73.47 |
| **Attune** | Kaptein et al., 2020 [23] | 2 years | 51.0 | 91.00 | 40.0 | 78.43 | NR | NR | NR | NR |
| | Hauer et al., 2020 [24] | 2 years | 51.5 | 92.60 | 41.0 | 79.80 | 44.6 | 85.0 | 41.0 | 90.58 |

NR = not reported, PR = patellar resurfaced, NPR = non-patellar resurfaced.

### 3.3.3. Comparison of HKSS

The pre-operative HSS value for the Freedom after an average follow-up of 5 years improve from $49.2 \pm 5.7$ to $88.8 \pm 3.4$, with a mean improvement of 39.6 points in the score. For the Persona after an average follow-up of 2 years improve from $46 \pm 10.5$ to $91 \pm 6.8$, with a mean improvement of 45 points in the score (Table 5).

**Table 5.** HSS results.

| Model | Study (First Author) | Average Follow-Up Time | Average Pre-Op HSS Score | Average HSS Score after Follow-Up Time | Mean Improvement |
|---|---|---|---|---|---|
| **Freedom** | Durbhakula, 2019 [12] | 5 years | $49.2 \pm 5.7$ | $88.8 \pm 3.4$ | 39.6 |
| **Persona** | Kim, 2019 [15] | 2 years | $46 \pm 10.5$ | $91 \pm 6.8$ | 45.0 |

### 3.3.4. Comparison of WOMAC

After 2 years follow-up, the pre-operative WOMAC value for the AKS improve from 28 to 85, and the Attune got a score of 81.2 post-operative, no information reported about the value pre-operative (Table 6).

**Table 6.** WOMAC-index results.

| Model | Study (First Author) | Follow-Up Time (Years) | Average Pre-Op WOMAC-Index | After Follow-Up Time WOMAC-Index |
|---|---|---|---|---|
| **AKS** | Souza et al., 2014 [19] | 2 | 28 | 85.0 |
| **Attune** | Hamilton et al., 2015 [31] | 2 | NR | 81.2 |

NR = not reported.

## 4. Discussion

### 4.1. Revision Rate

As the low-cost models are relatively recent, there are no long-term studies yet available. Furthermore, as they are not widely used, the patient cohort in those studies is much smaller than the ones in traditional prostheses or in registries. Thus, the revision rate cannot really be compared amongst the different models, as the length of study is very different and number of patients too. Moreover, short term-study mostly consider post-op complications such as infection but don't see long-term effect such as aseptic loosening. Nevertheless, the available data showed not remarkably worse performances in terms of revision rate due to the use of the low-cost implant compared to the traditional ones (Table 2).

### 4.2. Range of Motion (ROM)

Differences related to the range of motion (ROM) in the second postoperative year were negligible; mean 125° for the Persona, 124° for the Legion and 128° for the Triathlon. In the longer postoperative times (3 and 5 years) the mean was 124° for the Freedom. (Table 3) Lower values were measured in the for the Attune model, 113°. (Figure 1) No articles with the ROM for the AKS where found.

### 4.3. Knee Society Knee (KSKS) and Function (KSFS) Scores

The Knee society knee (KSKS) and function (KSFS) scores, and the mean change between preoperative and postoperative scores were used to compare between all the models except the Freedom (Table 4).

In the earlier test similar values of KSKS were found. AKS model (6 months postoperative) got 88 with patellar resurfaced and 84 with non-patellar resurfaced in the study. The Legion got 81 after 1 year postoperative (Figure 2).

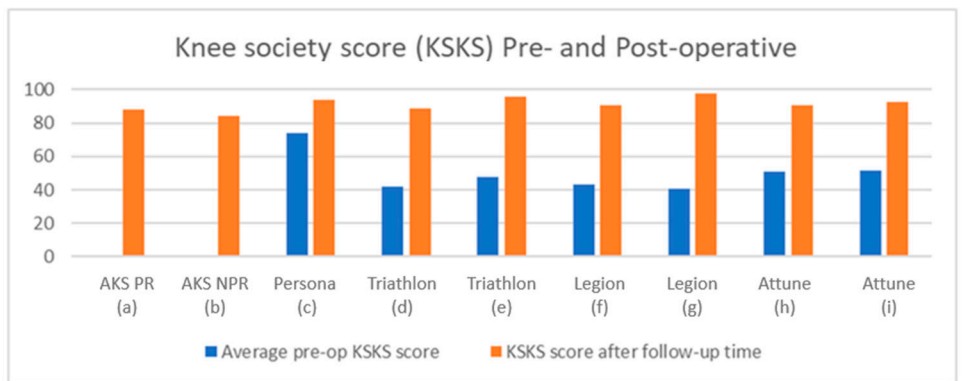

**Figure 2.** Knee society score (KSKS) Pre- and Post-operative for different TKA models, i.e., AKS PR (a [9]), AKS NPR (b [9]), Persona (c [19]), Triathlon (d [24], e [13]), Legion (f [27], g [4]), Attune (h [18], i [14]).

In later test, more than 2 years postoperative, we see and improvement in the values of KSKS in all the knee models, again, no notable differences were found. Higher postoperative values in KSKS were obtained with the Legion model after 5 years follow up (97.5). The rest of the values after two years follow up where 94 for the Persona, 89.5 for the Legion and 91.5 for the Attune (Figure 3).

The lowest values of KSFS were related to the Triathlon model, with 71.7 after two years follow-up and 85 post-operatively (Figure 4).

The mean change of 22 points between pre- and post-values where minor in the Triathlon model compared with the 38.5 and 41 points improvement of the Legion and Attune respectively. No significance for the 17 points improvement of the Persona model due to high a pre-operative value of 74 points. No KSFS results were found for the Freedom or AKS (Figure 5).

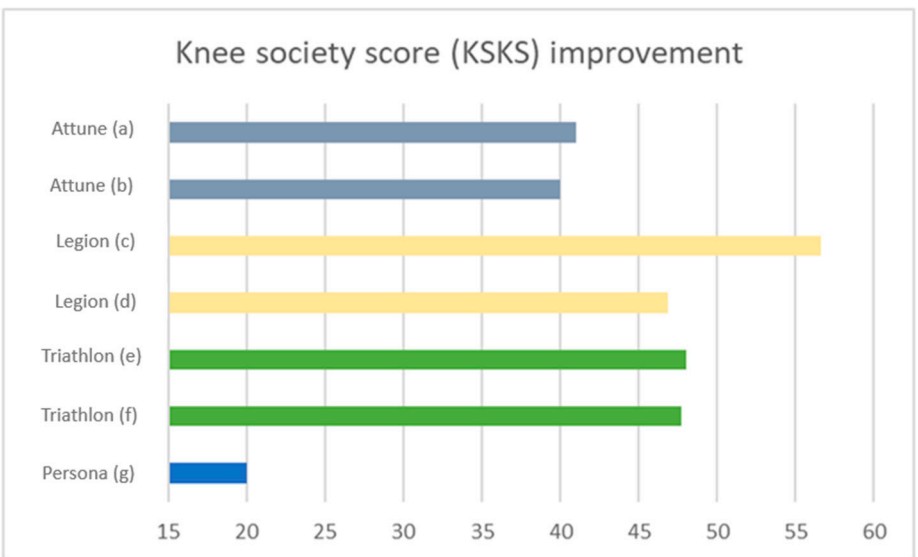

**Figure 3.** Knee society score (KSKS) improvement. Each color represents a specific TKA model, i.e., Attune (gray, a [14], b [18]), Legion (yellow, c [4], d [27]), Triathlon (green, e [13], f [24]), Persona (blue, g [19]).

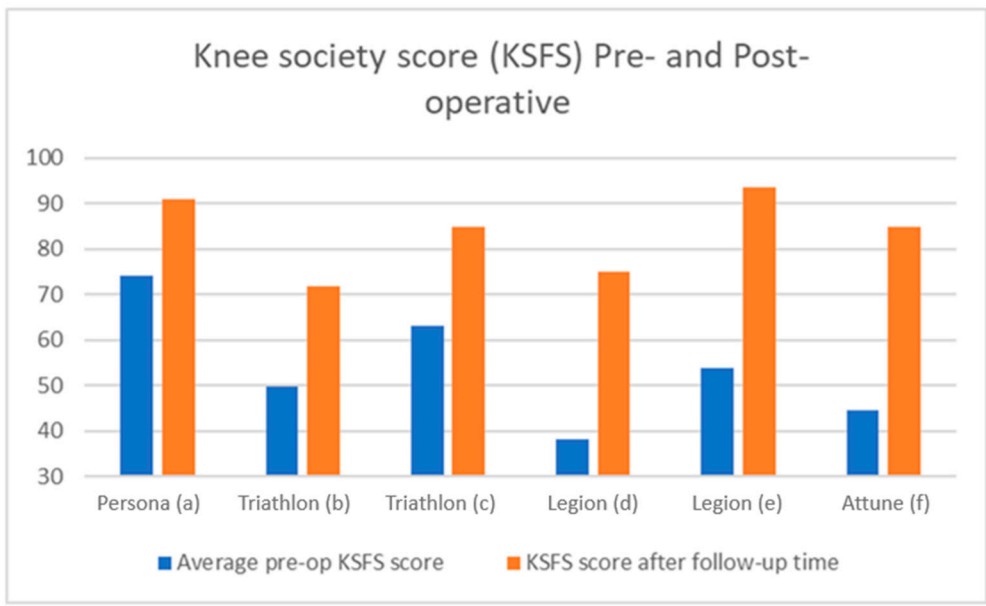

**Figure 4.** Knee society score (KSFS) Pre- and Post-operative for different TKA models, i.e., Persona (a [19]), Triathlon (b [24], c [13]), Legion (d [27], e [4]), Attune (f [14]).

*4.4. HSS and WOMAC Index*

To compare the clinical results with low-cost models we used the HSS and the WOMAC index.

The Hospital for special surgery (HSS) score were used to compare between the Freedom ($88.8 \pm 3.4$) and the Persona ($91 \pm 6.8$). The clinical outcomes scores improved compared with preoperative scores and were not significantly differences between the two models (Table 5).

The WOMAC index, after two years follow up, was used to compare between the Persona (85) and the Attune (81.2). The clinical outcomes scores improved compared with preoperative scores reaching similar results in the two models (Table 6).

Although the price difference is notable, the analysis of the selected studies shows there is no clear evidence of notably better performance in the traditional models. The results

show negligible differences relative to ROM, only slightly worse values in the HSS score and post-op KSKS score. However, WOMAC was better in the low-cost model. In general, the low-cost models performed as well, or not notably worse, than the traditional ones.

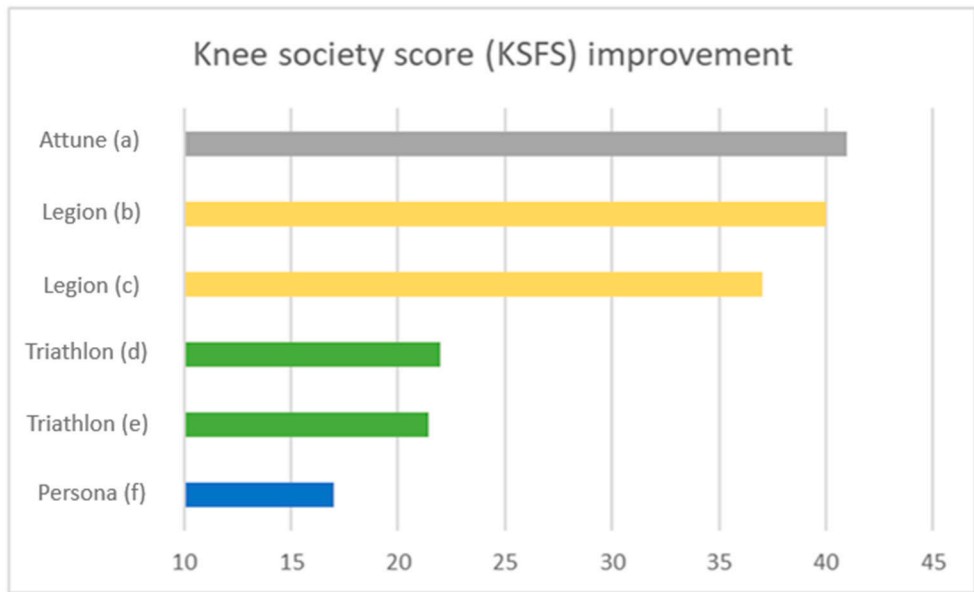

**Figure 5.** Knee society score (KSFS) improvement. Each color represents a specific TKA model, i.e., Attune (gray, a [14]), Legion (yellow, b [4], c [27]), Triathlon (green, d [13], e [24]), Persona (blue, f [19]).

*4.5. Limitations*

While the analysis of these results provides an initial understanding of the comparison between traditional and low-cost prostheses, it is important to acknowledge the limitations of this review. Firstly, the analysis was based on a small sample size, including only two low-cost models and four traditional models, which may not fully represent the diversity of the current total knee arthroplasty (TKA) market. Secondly, the comparison was limited to the results of only two or three studies for each model, which may not capture the complete picture. Additionally, it is worth noting that the low-cost models examined in this review are relatively new, and as such, there is a lack of long-term studies and survival rate data for these models. Therefore, a comparison of long-term outcomes between low-cost and traditional models is currently unavailable. These limitations emphasize the need for further research and a more comprehensive analysis to gain a deeper understanding of the performance and long-term durability of low-cost prostheses compared to traditional ones.

**5. Conclusions**

The objective of this paper was to evaluate whether there were clinical differences between low-cost prostheses sold in developing countries and traditional ones sold in the US and Europe. A literature review was conducted to identify clinical studies on both low-cost and traditional models. Four parameters, including range of motion, Knee Society Score (KSS), Hospital for Special Surgery (HSS) Score, and Western Ontario and McMaster Universities Arthritis Index (WOMAC), were analyzed for the selected models.

The review included 14 studies involving six different models, with data from 7138 knees meeting the inclusion criteria. The data utilized in the analysis spanned from six months to five years postoperatively. The comparison of the models showed that the low-cost models exhibited a similar range of motion improvement compared to traditional models. The improvement in HSS score was slightly smaller, the post-operative KSS was slightly lower but within the range of variation of traditional models, and the post-operative WOMAC score was better for the low-cost models. In summary, depending on

the parameter examined, the low-cost models performed equally well or only marginally worse than the traditional models.

However, it is important to note that this study has several limitations. It involved a small number of both low-cost and traditional models and a limited number of studies for each. To obtain a more comprehensive understanding of the comparison between low-cost and traditional prostheses, a more extensive study involving a larger variety of models and a greater number of patients should be conducted. Additionally, acquiring clinical results data for low-cost models may pose challenges, as there is currently limited accessible information available.

A more extensive study could potentially raise questions regarding the necessity of more expensive prostheses if they are found to perform similarly to low-cost alternatives. Conversely, it could also prompt further investigation into the factors contributing to the higher cost of traditional prostheses, such as materials, manufacturing processes, and tools, and their impact on overall prosthetic performance.

**Author Contributions:** Conceptualization, E.B. and B.I.; methodology, L.E.M. and C.D.; investigation, L.E.M. and C.D.; resources, L.E.M. and C.D.; data curation, E.B. and B.I.; writing—original draft preparation, L.E.M. and C.D.; writing—review and editing, E.B.; visualization, E.B.; supervision, B.I.; project administration, E.B. and B.I. All authors have read and agreed to the published version of the manuscript.

**Funding:** This research received no external funding.

**Institutional Review Board Statement:** Not applicable.

**Informed Consent Statement:** Not applicable.

**Data Availability Statement:** Data sharing not applicable.

**Conflicts of Interest:** The authors declare no conflict of interest.

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
