# Peer review of "Clinical Results of the Use of Low-Cost TKA Prosthesis in Low Budget Countries—A Narrative Review"

_prosthesis, doi:10.3390/prosthesis5030059_

Round 1

Reviewer 1 Report

This manuscript analyzes a limited number of publications to assess the performance of different low-cost and traditional prostheses. While the content is informative, this reviewer suggests addressing a few highlighted below.

- The title can be misleading depending on the readers. The manuscript is based on findings from other publications, but the title sounds like it is reporting a completely new study.

- The introduction heavily focuses on the prosthesis market in developing countries. Yet, this review is not sure how beneficial the discussion and conclusion would be to the prosthesis market in those countries. The introduction and the rest of the content are disjointed to this reviewer.

- The figures should be updated to be self-explanatory. For instance, the vertical axis of Figure 3 needs a unit, similarly the horizontal axis of Figure 4, etc. Also, are the numbers within the parenthesis represent reference numbers? If so, make it consistent with references in the content, i.e., use square brackets.

- As a review paper (sort of), the inference is severely lacking. The only takeaway message that is meaningful in conclusion is in lines 101~103.  Even this information is qualitative. Second half talks about the limitations of the current study, which further diminishes the impact of the manuscript.

Author Response

Comments and Suggestions for Authors

This manuscript analyzes a limited number of publications to assess the performance of different low-cost and traditional prostheses. While the content is informative, this reviewer suggests addressing a few highlighted below.

- The title can be misleading depending on the readers. The manuscript is based on findings from other publications, but the title sounds like it is reporting a completely new study.

The authors thank the reviewer for their suggestion, the title was modified accordingly in order to avoid misunderstandings.

- The introduction heavily focuses on the prosthesis market in developing countries. Yet, this review is not sure how beneficial the discussion and conclusion would be to the prosthesis market in those countries. The introduction and the rest of the content are disjointed to this reviewer.

The authors thank for the comment.

However, we consider that the primary purpose of the introduction is to provide a comprehensive rationale for conducting this review and to explain the significance of exploring knee prostheses in the context of BRICS countries, addressing the current state of knee prostheses in these regions (with all the eventual implications for healthcare providers, policymakers, and industry stakeholders) and how they consequently differ from the models available in the developed countries. While we acknowledge that the introduction may seem heavily focused on the prosthesis market in developing countries, we believe this emphasis is essential to set the stage for a comprehensive analysis of knee prostheses within the unique socioeconomic and healthcare landscapes of these countries.

- The figures should be updated to be self-explanatory. For instance, the vertical axis of Figure 3 needs a unit, similarly the horizontal axis of Figure 4, etc. Also, are the numbers within the parenthesis represent reference numbers? If so, make it consistent with references in the content, i.e., use square brackets.

The figures were modified accordingly to the suggestion, adding the unit of measurement where possible (scores are indeed a-dimensional, therefore no unit was added). The references’ brackets were also modified to be square, in according to the manuscript format.

- As a review paper (sort of), the inference is severely lacking. The only takeaway message that is meaningful in conclusion is in lines 101~103.  Even this information is qualitative. Second half talks about the limitations of the current study, which further diminishes the impact of the manuscript.

In response to the comment concerning the message in lines 101~103, we would like to clarify that these lines are actually part of the "Material & Methods" section. We believe there might be some confusion, as the actual conclusion section is located at a different part of the paper.

Concerning the manuscript's impact, our objective is to consolidate and assess the existing literature on knee prostheses in low budget countries. Given the scarcity of data available in this particular context, the limited number of studies presents an inherent challenge in conducting comprehensive reviews in certain areas. By acknowledging these limitations, our goal is to ensure transparency and provide readers with the necessary context, avoiding any exaggeration of findings while recognizing the boundaries within which our conclusions are drawn.

Reviewer 2 Report

The subject matter of the manuscript is potentially interesting. It has many limitations, which are recognized by the authors. I would suggest publication as a Review rather than an Original Article.

Some notes:

- next to the companies mentioned, indicate the city and nationality of the headquarters in brackets.

- whenever Zimmer is mentioned, write Zimmer Biomet

- line 104, at the end of the names of the prosthesis models write models

- table 2, write in the caption what CS and PS mean

- paragraph 3.3 useless, contains information already mentioned

- in all the figures, indicate the unit of measurement and explain in the caption what the numbers in brackets next to the prosthesis models mean

table 3 redundant with figure 1

Author Response

Comments and Suggestions for Authors

The subject matter of the manuscript is potentially interesting. It has many limitations, which are recognized by the authors. I would suggest publication as a Review rather than an Original Article.

The authors thank the reviewer for their suggestion, the title and rationale were modified accordingly, in order to avoid misunderstandings.

Some notes:

- next to the companies mentioned, indicate the city and nationality of the headquarters in brackets.

- whenever Zimmer is mentioned, write Zimmer Biomet

- line 104, at the end of the names of the prosthesis models write models

- table 2, write in the caption what CS and PS mean

The manuscript was modified accordingly to the suggestions and the due clarifications were added.

- paragraph 3.3 useless, contains information already mentioned

The paragraph mentioned was modified to reduce redundancy and improve readability.

- in all the figures, indicate the unit of measurement and explain in the caption what the numbers in brackets next to the prosthesis models mean

The figures were all revised in order to be compliant with the manuscript format.

table 3 redundant with figure 1

The authors agree with the reviewer’s comment; therefore, Figure 1 was removed to avoid redundancy.

Reviewer 3 Report

Thank you for your submission. My overall impression is that it is too long. The manuscript could be improved with a clearly defined time frame of the literature searched and of which databases. Then clearly defined research questions with results, discussion, and conclusions based upon your questions. Clearly, there are untapped markets that could be conveyed in fewer words than currently worded in the introduction. 

Author Response

Comments and Suggestions for Authors

Thank you for your submission. My overall impression is that it is too long. The manuscript could be improved with a clearly defined time frame of the literature searched and of which databases. Then clearly defined research questions with results, discussion, and conclusions based upon your questions. Clearly, there are untapped markets that could be conveyed in fewer words than currently worded in the introduction. 

The authors thank the reviewer for their comments; we conducted a thorough search of the available literature on knee prostheses in BRICS countries and compiled all relevant findings. Due to the limited number of studies found, we have explicitly highlighted the limitations in the study, and we agree that these limitations are essential to address. We wish to highlight our emphasis on developing and low budget countries, as we relied on the scarce scientific references relevant to them, and the fact that even more limited scientific references were available for other regions, therefore we decided to concentrate our discussion on the data available for the BRICS countries.

Reviewer 4 Report

This is an interesting review, however, can benefit from small revision and further work defining methods. 

Consider relabelling the title of the paper to be more clear that this is a systematic review, and consider meta-analysis.  Low budget is different than BRICs countries, different than developing countries.

The definition of both low-cost and the methods selection for models in question need to be defined. Please consider a definition for how models were determined, and if there are any other models that should be in scope. 

Why were the BRIC countries used - could consider expanding to BRICS countries? 

Some of the numbers in Table 1, including mean / weighted average mean are concerning - please review in detail. Sample size numbers seem to be off, e.g. Legion and Atune avg totals? Consider WAVG vs raw AVG

English language is appropriate, a quick re-review of grammar and spelling is merited. 

Author Response

Comments and Suggestions for Authors

This is an interesting review, however, can benefit from small revision and further work defining methods. 

Consider relabelling the title of the paper to be more clear that this is a systematic review, and consider meta-analysis.  Low budget is different than BRICs countries, different than developing countries.

The authors thank the reviewer for their suggestion, the title was modified mentioning that it is a narrative review in order to avoid misunderstandings. Concerning the differences among low budget and BRICS countries, the authors wanted to highlight the fact that BRICS countries were considered as developing countries in this study as a consequence of the economic availability applied to the field of orthopedics, with all the limitations mentioned. The fact of being part of the BRICS is then of secondary importance, with the purely economic constraint being the main representative factor in order to be taken into account for this review. The text was modified in order to improve clarity and avoid misunderstandings, by mentioning the BRICS only as an introduction and then focusing on the term of “low budget” and “developing countries”.

The definition of both low-cost and the methods selection for models in question need to be defined. Please consider a definition for how models were determined, and if there are any other models that should be in scope. Why were the BRIC countries used - could consider expanding to BRICS countries? 

We have carefully considered your suggestion regarding expanding the terminology to BRICS (Brazil, Russia, India, China, and South Africa), and we modified the manuscript accordingly; however, as mentioned in the introduction, the main focus is on the “low budget” itself, represented by the lower costs which the orthopedic companies have to offer in order to be affordable in the countries addressed. The main challenge we then encountered during our review is the lack of extensive literature concerning the specific topic of knee prostheses: due to this limitation, we opted to conduct a narrative review, allowing us to gather and synthesize all available information on the subject while still mentioning the limitations of this approach.

We acknowledge that in the future, there may be emerging research and a greater availability of information in this area: with the arrival of more sources in the literature, we will undoubtedly be interested in exploring new approaches and methodologies to delve deeper into the topic.

Some of the numbers in Table 1, including mean / weighted average mean are concerning - please review in detail. Sample size numbers seem to be off, e.g. Legion and Atune avg totals? Consider WAVG vs raw AVG

The authors indeed agree with the reviewer, the Table reported incorrect values due to typos in the programming of the relative excel file. The Tables were thoroughly revised in order to solve this issue and the new versions are available in the revised manuscript.

Round 2

Reviewer 4 Report

Thank for carefully reviewing comments and addressing them. The updated version, with updated tables and results appear consistent with the methodology outlined. The authors could consider a scoping review rather than a formal systematic review based on the comments they have provided, but this could be afuture work. 

Minor edits could be made to grammatical structure and wording. For example, in tables the words “percentage of improvement” or “improvement percent” are more appropriate than “percentual improval”